# Study on the Seismic Performance of Stiffened Corrugated Steel Plate Shear Walls with Atmospheric Corrosion

**DOI:** 10.3390/ma15144920

**Published:** 2022-07-14

**Authors:** Xiaoming Ma, Yi Hu, Liqiang Jiang, Lizhong Jiang, Guibo Nie, Hong Zheng

**Affiliations:** 1Key Laboratory of Earthquake Engineering and Engineering Vibration, Institute of Engineering Mechanics, China Earthquake Administration, Harbin 150080, China; 2020228041@chd.edu.cn (X.M.); nieguibo0323@163.com (G.N.); 2Key Laboratory of Earthquake Disaster Mitigation, Ministry of Emergency Management, Harbin 150080, China; 3School of Civil Engineering, Chang’an University, Xi’an 710061, China; cehzheng@chd.edu.cn; 4School of Civil Engineering, Central South University of Forestry and Technology, Changsha 410004, China; hyi_1991@163.com; 5School of Civil Engineering, Central South University, Changsha 410075, China; lzhjiang@csu.edu.cn

**Keywords:** corrugated steel plate shear wall (CSPW), finite element method (FEM), lateral performance, hysteretic performance, atmospheric corrosion

## Abstract

Corrugated steel plate shear walls (CSPWs) with three different stiffening methods are proposed in this paper, including unstiffened CSPWs (USWs), cross stiffened CSPWs (CSWs) and asymmetric diagonal-stiffened CSPWs (ASWs). A numerical model was established by ABAQUS 6.13 based on the validation of an existing cyclic test on a CSPW. This paper presents an investigation of the lateral performance under monotonic loading, seismic performance under cyclic loading and seismic performance under atmospheric corrosion of USW, CSW and ASW. The results show that (1) Stiffeners can improve the elastic critical buckling load, the initial stiffness and the ultimate shear resistance of CSPWs, and the effect of asymmetric diagonal stiffeners is more significant than that of cross stiffeners; (2) Stiffeners can improve the energy dissipation capacity and ductility, delay stiffness degradation and reduce the out-of-plane deformation of CSPWs, and the hysteretic performance of ASWs is obviously better than that of CSWs; and (3) Under atmospheric corrosion, stiffeners are conducive to inhibiting buckling and improving the seismic performance of CSPWs, while the seismic performance of CSWs is significantly affected by corrosion, so asymmetric diagonal stiffeners are better than cross stiffeners in improving the seismic performance of CSPWs. Meanwhile, the formula of ultimate shear resistance of corroded specimens is also fitted in this paper, which can provide design suggestions for practical engineering.

## 1. Introduction

With lighter weight, higher strength and better ductility, the damage degree of steel structures is much lower than that of reinforced concrete structures after earthquakes un-der the same conditions. To address problems existing in traditional structures, the steel plate shear wall (SPSW) has been proposed. The SPSW is a kind of lateral force resistant structural system that emerged in the 1970s. Due to its high initial stiffness, good ductility and seismic performance, simple connection and reduced structural deformation, the SPSW has been widely used in the lateral force resisting system of middle- and high-rise buildings [1,2,3,4]. However, the steel plate is prone to buckling, which can cause a reduction in the shear resistance and lateral stiffness. Meanwhile, the steel plate has poor energy dissipation capacity, and it can also form a tension field, which is disadvantageous to the boundary column [5,6,7,8].

To solve the problem that the steel plate is prone to buckling, many experiments and theoretical analyses were conducted. Berman and Bruneau [9,10,11] proposed the strip mod-el of the flat steel plate, and the formulae for calculating the ultimate shear resistance of the SPSW when the infilled steel plate is hinged and rigidly connected to the boundary frame are proposed, which were verified by experiments. Studies have shown that stiffeners can significantly improve the energy dissipation capacity, ultimate shear resistance, initial stiffness and ductility [12]. Nie et al. [13] studied the influence of 6 kinds of stiffeners on the SPSW, and the results showed that among the six kinds of stiffeners, the asymmetric diagonal stiffened SPSW had the largest critical buckling load, followed by the cross stiffener. According to Cao et al. [14,15], compared with unreinforced SPSWs, the new SPSW with X-shaped restrainers has been improved in some respects, including the shear resistance, peak load, energy dissipation capacity and out-of-plane displacement constrained efficiency, and its hysteretic curve tends to be full. Khaloo et al. [16] conducted numerical investigations on a diagonal stiffened SPSW, and the results showed that the stiffened SPSW showed good crack resistance in both the elastic and plastic stages.

CSPWs have attracted more attention than traditional SPSWs because CSPWs exhibit higher out-of-plane stiffness, shear resistance, ductility and energy dissipation capacity [17,18]. Emami et al. [19,20] conducted numerical investigations and experiments on the cycle performance of CSPWs and SPSWs, and the results showed that compared to SPSWs, CSPWs had higher out-of-plane stiffness and buckling strength and a fuller hysteretic curve. Meanwhile, the beam-only connected SPSW was proposed first to avoid high-flexural and axial demands in the boundary column due to the tension field [21]. Paslar et al. [22] established 57 numerical models to investigate the connection between steel plates and boundary frames. Elastic buckling can be avoided by reasonable design of the corrugated size, but there is a certain relationship between the ultimate shear resistance and the geometric parameters of the CSPW, so geometric parameters should be reasonably selected to avoid a significant decrease in the post-buckling strength [23,24,25,26,27,28]. To inhibit the out-of-plane buckling of corrugated steel plates, corrugated steel plates with stiffeners have been proposed by researchers. Tong [29] proposed three new types of CSPWs and then conducted experimental, numerical and theoretical investigations. The results showed that stiffeners can effectively restrain the out-of-plane buckling of CSPWs and improve the ultimate shear resistance and energy consumption of CSPWs to some extent. Wang [30] and Zheng et al. [31,32] proposed a new type of cross stiffened CSPW and established FEM models. Then, the lateral resistance, seismic performance and the influence of the dimensions of the CSPW on the mechanical performance were analysed. The results showed that in the middle and late loading stage of the CSPW, the steel plate can have in-elastic buckling, resulting in the decline of the ultimate shear resistance and the final failure. The cross stiffener can play a good role in inhibiting buckling and further improving the mechanical performance of the CSPW.

Meanwhile, corrosion is a problem for a large number of steel structures. In coastal areas, corrosion is an important factor leading to age-related structural degradation of steel structures; it can reduce the strength of steel, cause stress concentration, and finally reduce the bearing capacity of structures [33,34]. Zheng et al. [35,36,37] conducted accelerated corrosion tests and quasi-static cyclic tests on steel frames and joints, and Xu et al. [38,39] conducted accelerated corrosion tests, quasi-static cyclic tests and finite element analyses on seven H-shaped steel columns. The results showed that with increasing corrosion degree, the forming of the plastic hinge was accelerated, and the ultimate lateral resistance, stiffness and energy dissipation capacity of the specimens decreased significantly.

Although many experimental and theoretical investigations have been conducted on CSPWs with stiffeners, the current research mainly compares the mechanical properties of CSPWs before and after stiffening, and research comparing CSPWs with different stiffening methods is still insufficient. Meanwhile, although many experimental and numerical investigations of steel structures under atmospheric corrosion have been conducted [35,36,37,38,39], there are still few investigations on the seismic performance of corroded CSPWs [40,41,42]. Therefore, this paper carries out finite element analyses on CSPWs with different stiffening methods by using ABAQUS. The lateral performance, seismic performance and hysteretic performance under atmospheric corrosion are studied, and the analysis results can provide practical suggestions for practical engineering design.

## 2. Numerical Modelling

### 2.1. Model Description

This paper proposes three types of CSPWs with different stiffening methods, including USW, CSW and ASW. According to the Chinese standards of GB50017-2017 [43], JGJ/T380-2015 [44] and GB50011-2010 [45], the dimensions of the USW, CSW and ASW are presented in Figure 1. The specimen consists of the boundary frame, infilled corrugated steel plate and the stiffener. The section of the boundary column is H 400 × 300 × 18 × 22. The top-beam section and the bottom-beam section are designed as the same sizes, and they are H 350 × 300 × 14 × 20. The section of the middle beam is H 300 × 250 × 12 × 16. The beam column is rigidly connected, and the stiffeners are welded in the node domain. The dimensions of the infilled corrugated steel plate are presented in Figure 1. Vertical trapezoidal corrugated steel plates are used for these specimens, and the thickness is 6 mm, the wave height is 90 mm, the wavelength is 300 mm, and the CSPW is only welded with the frame beam. The stiffeners are welded to the surface of the corrugated steel plate. The dimensions of the cross stiffener of the CSW are −3000 × 70 × 7. The stiffeners are arranged asymmetrically on the front and back sides of the corrugated steel plate, and the dimensions of the long stiffener and short stiffener are −4240 × 70 × 7 and −2120 × 70 × 7, respectively. Q355 steel is used for the boundary frame and stiffeners, and Q235 steel is used for the corrugated steel plate.

Numerical models of USW, CSW and ASW are shown in Figure 2. In the numerical models, S4R four-node shell elements are used for modelling the boundary frame, the corrugated steel plates and the stiffeners. The maximum size of the elements is 70 mm for steel, and the mesh sensitivity analysis reveals that the mesh size is dense enough for reasonable accuracy and computation time. The constitutive relation of the steel is a bilinear kinematic model; before the steel reaches the yield strength, the slope is *E*, and after reaching the yield strength, the slope is 0.03 *E*. A “Merge” combination command is used to simulate the weld connections in the beams and the columns. A “Tie” command is used to connect the corrugated steel plates to the boundary beams and stiffeners. Initial defects are introduced into the model by modifying the keyword, and the amplitude is taken as 1/1000 [46].

The lateral loads were imposed using the displacement-control scheme, as shown in Figure 3. Loading steps were integer multiples of yield displacement *δ*_y_. The spacing between each control point was *δ*_y_, and it could be known when an inflection appeared in the curve in Figure 3, repeated three times at each control point.

### 2.2. Validation of FEM

Currently, no existing cyclic tests have been performed to validate the FEM model of CSWs and ASWs. This paper selects test results from specimen S-1 from the Ref. [47]. S-1 is shown in Figure 4. The thickness, height and width of the corrugated steel plates were 3 mm, 1 m and 1.75 m, respectively. Q345 steel was used for the boundary frame of S-1, and Q235 steel was used for the infilled corrugated steel plate of S-1. The numerical model of specimen S-1 is then built using the same modelling method presented in Section 2.1, and the material properties of the test specimen are shown in Table 1.

The comparisons of the hysteretic curves and envelope curves obtained from FEM analyses and tests are shown in Figure 5, and the detailed results are shown in Table 2. The hysteretic curves of FEM analyses and tests were full, and the overall trend of envelope curves from FEM analyses and tests was basically consistent. The error of yield strength between FEM analyses and tests is only 0.7%. The yield displacement and peak load of the tests are higher than those of FEM analyses, and the errors are 8.8% and 3.8%, respectively.

The comparisons of the failure modes of S-1 between FEM analyses and tests are shown in Figure 6. The results of the comparisons show that the FEM model accurately simulates the buckling modes and buckling positions of boundary columns and corrugated steel plates. It can be found that the numerical model in Section 2.1 is reasonable, and it can be used for the following study.

## 3. Comparative Study of the Lateral Resistance of the CSPW

FEM models of USW, CSW and ASW are built as shown in Figure 7 and their lateral resistance under monotonic loading is compared and analyzed. The modelling method is similar to that in Section 2.1, and a “Tie” combination command is used to simulate the weld connections in the stiffener and the corrugated steel plate. The constitutive relation of the material is linear elasticity. The boundary condition of the corrugated steel plate is that the upper and lower sides are hinged with the boundary frame. Axial forces *P*_1_ and *P*_2_ are applied at two coupling points on the top of the boundary columns to simulate the vertical load, such as gravity, and *P*_1_ = *P*_2_ = 600 kN. Then, horizontal load is applied at the coupling point on the height of the centerline of the top beam until the specimen fails.

### 3.1. Comparison of Monotonic Load–Displacement Curves

The load–displacement curves of the USW, CSW and ASW are shown in Figure 8, and the detailed results are shown in Table 3. Due to stiffeners, the properties of three types of specimens such as the initial stiffness, yield load and ultimate shear resistance are different. The yield load and peak load of ASW were the largest, followed by CSW. With increasing lateral displacement, USW showed the earliest decline period with a large decline range, and ASW showed the last decline period with a gentler decline. This means that ASW has the best ductility and USW has the worst ductility.

Compared to USW, the initial lateral stiffness, yield load, yield displacement, peak load and displacement at the peak load of the CSW increase by 6.05%, 7.31%, 1.44%, 7.48% and 39.84%, respectively. Compared to the CSW, the initial lateral stiffness, yield load, yield displacement, peak load and displacement at the peak load of the ASW increase by 5.1%, 5.8%, 6.02%, 8.7% and 40%, respectively. Therefore, cross stiffeners can effectively improve the lateral resistance of the CSPW, and asymmetric diagonal stiffeners can be more effective than cross stiffeners. Meanwhile, asymmetric diagonal stiffeners can also avoid the stress concentration and complex connection caused by diagonal stiffeners on the same side.

### 3.2. Comparison of Out-of-Plane Deformations

The out-of-deformations of the USW, CSW and ASW at failure under monotonic loading are shown in Figure 9. USW had serious out-of-plane deformation and buckling. Due to the inhibition of cross stiffeners, the out-of-plane deformation of the CSW was low-er than that of the USW. The deformation of ASW at failure was mainly local deformation, and the out-of-plane deformation of ASW was the lowest among them. The maximum out-of-plane deformation of USW, CSW and ASW is 231.7 mm, 145.61 mm and 88.01 mm, respectively, so compared to ASW, the maximum out-of-plane deformation of CSW and USW increases 39.56% and 62.02%, respectively. It is revealed that asymmetric diagonal stiffeners can inhibit the out-of-plane deformation of the CSPW more effectively than cross stiffeners.

### 3.3. The Comparison of Stress Distributions

The stress distributions of USW, CSW and ASW at failure are shown in Figure 10a–f. When USW was damaged, the corrugated steel plate buckled seriously, full section yielding occurred at the bottom of the boundary column, and plastic buckling occurred at the end of the middle beam. Due to the inhibition of cross stiffeners, the lateral displacement of the CSW at buckling was larger than that of the USW, the plastic hinge at the bottom of the boundary column was formed later, and the buckling of the corrugated steel plate of the CSW at failure was slighter than that of the USW. Due to the inhibition of asymmetric diagonal stiffeners, the corrugated steel plate of ASW had large in-plane stiffness, the plastic area at the bottom of the boundary column of ASW was smaller than that of CSW, and when ASW was damaged, there was no full section yielding at the bottom of the boundary column.

The comparative results show that due to the inhibition of cross stiffeners and asymmetric diagonal stiffeners, the plastic hinge at the bottom of the boundary column is formed later, and the failure of the whole specimen is delayed. Meanwhile, asymmetric diagonal stiffeners are better than cross stiffeners in inhibiting the inelastic buckling in-stability of CSPWs.

## 4. Comparative Study on Seismic Performance of the CSPW

Finite element analyses of USW, CSW and ASW under cyclic loads are carried out, and the modelling method is the same as that in Section 3. A vertical load is applied at two coupling points on the top of the boundary columns, and then a horizontal load is applied at the coupling point on the beam-column joint.

### 4.1. Comparison of Cyclic Load–Displacement Curves

The hysteretic curves and the envelope curves of the specimens are shown in Figure 11, and detailed results are shown in Table 4. Compared to USW and CAW, the hysteresis loop of specimen ASW was fuller. With global buckling of the CSPW, the lateral stiffness of the specimens decreased, and the stiffness of the CSW and ASW decreased slightly in the late loading stage. With increasing lateral displacement, the specimen yielded and gradually entered the stage of plastic development, resulting in loss of stiffness. Then, the ultimate shear resistance of the specimen decreased significantly, and the specimen entered the stage of plastic failure. Compared to USW, the initial stiffness, yield load, yield displacement, peak load and ductility of the CSW increase by 4.1%, 6.82%, 11.35, 6.7% and 27.64%, respectively, and the initial stiffness, yield load, yield displacement, peak load and ductility of the ASW increase by 12.15%, 11.66%, 15.32%, 14.4% and 46.4%, respectively. Therefore, asymmetric diagonal stiffeners can inhibit the buckling of CSPWs and improve hysteretic performance more effectively than cross stiffeners.

### 4.2. Comparison of Energy Dissipation Capacity

The energy dissipations of the specimens are shown in Figure 12. The energy dissipation of the ASW under each loading step was higher than that of the USW and CSW. After the loading step reached 2δy, the energy dissipation of the ASW increased the fastest, followed by that of the CSW. The maximum values of energy dissipation of USW, CSW and ASW are 1,207,511 kN·mm, 1,476,870 kN·mm and 1,865,640 kN·mm, respectively. Compared to USW and CSW, the energy dissipation of ASW improves 26% and 54%, respectively. Therefore, stiffeners can improve the energy dissipation of CSPWs, and asymmetric diagonal stiffeners can improve the energy dissipation of CSPWs more effectively than cross stiffeners.

### 4.3. Comparison of the Degradation of the Stiffness and Strength

The curves of stiffness degradation of USW, CSW and ASW are shown in Figure 13. The stiffness degradation of the specimens had the same trend. The stiffness of the ASW at each loading step was higher than that of the USW and CSW. After reaching the peak load, the stiffness degradation of ASW was smaller than that of USW and CSW. Therefore, asymmetric diagonal stiffeners can better inhibit buckling and improve the in-plane stiff-ness of CSPWs at the late loading stage.

The strength degradation coefficient (*λ_i_*, *i* = 1, 2, ……) is an index to evaluate the strength degradation of the structure, and the calculation formula is:(1)$$\lambda_{i}=\frac{F_{j}^{i+1}}{F_{j}^{i}},$$
where $F_{j}^{i+1}$ is the peak load of cycle *i* + 1 at the *j*th loading step and $F_{j}^{i}$ is the peak load of cycle *i* at the *j*th loading step.

The variation trend of *λ*_1_ and *λ*_2_ with loading displacement is shown in Figure 14. Before reaching the peak load, *λ_i_* of the USW, CSW and ASW is approximately about 1, and after reaching the peak load, *λ_i_* of the USW, CSW and ASW decreases by 6%, 4.5% and 2%, respectively, and *λ_i_* of the ASW is not less than 0.98. Compared to USW and CSW, *λ_i_* of ASW increases by a maximum of 4.8% and 3.7%, respectively, so stiffeners can improve the stability of ultimate shear resistance to some extent, but the improvement is not as obvious.

## 5. Comparative Study of the Seismic Performance of CSPWs under Atmospheric Corrosion

FEM models of USW, CSW and ASW are built under different corrosion levels, and the modelling method is the same as that in Section 3. Then, the hysteretic performance of the specimens is compared and analyzed.

### 5.1. Material Properties after Corrosion

To achieve the long-term corrosion effect of steel in a chloride environment for a short time, the indoor or outdoor accelerated corrosion testing method is generally adopted. The outdoor accelerated corrosion test and results from the literature [35] were selected in this paper. The NaCl solution concentration was 50 ± 5 g/L, and the pH value was 6.5–7.2. The fitting formula from the literature [35] was selected to calculate the yield strength of corroded steel, as shown in Equation (2).
(2)$$f_{y}=(1-0.902\times \frac{1-E/E_{0}}{0.897})f_{y0},$$
where *f*_y0_ and *E*_0_ are the yield strength and elastic modulus of uncorroded steel, respectively, and *f*_y_ and *E* are the yield strength and elastic modulus of corroded steel, respectively. *f*_y0_ of steel are 235 MPa and 355 MPa, respectively, and *E*_0_ is 206,000 MPa.

The time of the accelerated corrosion test is converted to the time of actual outdoor corrosion. The rate of accelerated corrosion is 1208 μm·a^−1^ in this paper, and the rate of atmospheric corrosion in Beijing is 11.7 μm·a^−1^ [48,49]. These two rates are used for conversion, and the results are shown in Table 5. The change in the elastic modulus of steel during the corrosion process is shown in Figure 15. To better reflect the changes in material properties during the corrosion process, the elastic modulus after 20, 60 and 120 days of corrosion [35], which is actual corrosion of 5.657, 16.972 and 33.945 years, are selected for FEM models, as shown in Table 6.

### 5.2. Comparison of the Hysteretic Performance of Corroded CSPWs

Load–displacement curves of corroded specimens for 20, 60 and 120 days are shown in Figure 16a–f, and the detailed results are shown in Table 7. Under different corrosion levels, the hysteretic loop of ASW was the fullest among them, and when corroded for 20 and 60 days, the stiffness and ultimate shear resistance of CSW and ASW decreased more gently than that of USW, but when corroded for 120 days, the trends of ultimate shear resistance degradation and stiffness degradation of USW and CSW were basically consistent, and the stiffness and ultimate shear resistance of ASW decreased more gently than that of USW and CSW.

When the corrosion time is 20 days, compared to USW, the initial stiffness, yield load, peak load and ductility factor of the CSW increase by 2.16%, 17.12%, 7.8% and 25.3%, respectively; the initial stiffness, yield load, peak load and ductility factor of ASW increase by 11.49%, 25.41%, 15.69% and 46.13%, respectively. When the corrosion time is 60 days, compared to USW, the initial stiffness, yield load, peak load and ductility factor of the CSW increase by 2.33%, 15.85%, 4.32% and 9.04%, respectively; the initial stiffness, yield load, peak load and ductility factor of ASW increase by 5.81%, 25.28%, 13.43% and 36.7%, respectively. When the corrosion time is 120 days, compared to USW, the initial stiffness, yield load, peak load and ductility factor of the CSW increase by 1.87%, 13.65%, 2.11% and 10.42%, respectively; the initial stiffness, yield load, peak load and ductility factor of the ASW increase by 6.26%, 18.76%, 4.52% and 39.49%, respectively.

Therefore, under different corrosion levels, stiffeners can improve the initial stiffness, yield load and ductility of CSPWs, increase the energy dissipation capacity and delay the stiffness degradation of CSPWs, so stiffeners can improve the hysteretic performance of CSPWs. However, with increasing corrosion time, the hysteretic performance of CSWs is affected more obviously by corrosion than that of USWs and ASWs, so asymmetric diagonal stiffeners can improve the hysteretic performance of corroded CSPWs more effectively than cross stiffeners.

### 5.3. Comparison of the Hysteretic Performance of CSPWs before and after Corrosion

Load–displacement curves of specimens before and after corrosion are shown in Figure 17. With increasing corrosion time, notable pinching was observed for the hysteretic loops of corroded CSWs, and the trend of stiffness degradation for corroded USW was basically consistent. Compared to the uncorroded CSW, the initial stiffness of USW after corrosion for 20, 60, and 120 days decreases by 1.78%, 4.13% and 6.53%, respectively; the peak load decreases by 2.73%, 4.14% and 4.91%, respectively; and the ductility factor de-creases by 2.68%, 5.53% and 8.38%, respectively.

With increasing corrosion time, the hysteresis curve of the corroded CSWs was pinched, and the CSW stiffness degradation trend of the uncorroded CSWs corroded for 20 and 60 days was basically consistent, but the corroded CSWs corroded for 120 days had a significant decrease in stiffness and shear resistance. Compared to the uncorroded CSW, the initial stiffness of USW after corrosion for 20, 60, and 120 days decreases by 3.6%, 5.77% and 8.53%, respectively; the peak load decreases by 6.85%, 11.35% and 13.93%, respectively; and the ductility factor decreases by 2.86%, 11.78% and 12.7%, respectively.

With increasing corrosion time, the hysteresis curve of corroded ASW was pinched more gently than that of USW and CSW, and the trend of stiffness degradation for corroded ASW was basically consistent. Compared to uncorroded ASW, the initial stiffness of ASW after corrosion for 20, 60, and 120 days decreases by 2.37%, 9.56% and 11.44%, respectively; the peak load decreases by 1.63%, 4.94% and 13.13%, respectively; and the ductility factor decreases by 2.86%, 11.78% and 12.7%, respectively.

Therefore, corrosion reduces the initial stiffness, ultimate shear resistance, ductility and energy dissipation capacity of USW, CSW and ASW. Due to the corrosion of stiffeners, the effect of stiffeners on increasing initial stiffness, yield load and delaying stiffness degradation has been reduced significantly, so the indexes of corroded CSPWs with stiffeners including CSW and ASW decreased more significantly than those of USW with increasing corrosion time, and the indexes of USW after corrosion decreased slowly with increasing corrosion time, revealing that corrosion has little effect on the hysteretic performance of USW. Meanwhile, the ultimate shear resistance and stiffness of ASW after corrosion decreased gently, and the hysteretic curve was still full, so it still has good seismic performance.

### 5.4. The Fitting Formulae of Ultimate Shear Resistance

The time of the accelerated corrosion test is converted to the time of actual outdoor corrosion. From Section 5.1, the times of the corrosion test are 20, 60 and 120 days, and the corresponding times of actual corrosion are 5.657, 16.972 and 33.945 years, respectively. The variation in the ultimate shear resistance of the specimens with the actual corrosion time is shown in Figure 18. The finite element results are selected to fit the formulae of ultimate shear resistance of specimens, as shown in Equations (3)–(5).
(3)$$y=102.809e^{-x/7.345}+2041.223,$$
(4)$$y=296.377e^{-x/26.38}+1997.899,$$
(5)$$y=-186.533e^{x/34.021}+2637.706,$$

Then FEM results of corroded specimens for 40 and 80 days are used to validate the proposed Formulas (3)–(5). According to Table 5, the actual corrosion time of specimens for outdoor accelerated corrosion time 40 and 80 days are 11.315 and 22.63 years, respectively. From the Ref. [35], the elastic modulus of corroded specimens for 40 and 80 days are 199,348 MPa and 190,493 MPa, respectively. The comparative results of ultimate shear resistance between FE analysis and proposed formulae are shown in Table 8. The ultimate shear resistance results based on the proposed theoretical model are very closed to the FEM results, and the maximum error between theoretical results and FEM results is 1.9%. The comparisons indicate that the proposed fitting formulae could effectively predict the ultimate shear resistance of corroded CSPWs.

## 6. Conclusions

In this research, CSPWs with three different stiffening methods are proposed. The in-fluence of different stiffening methods on the lateral and seismic performance of CSPWs before and after corrosion is examined. The conclusions are as follows:(1)The FEM model of specimen S-1 from the Ref. [47] is established, and test results from S-1 are selected to validate the FEM model. The results show that the initial stiffness and ultimate shear resistance analyzed by FEM are basically consistent with the test results, and the maximum error is less than 10%.(2)Stiffeners can effectively improve the initial stiffness, ultimate shear resistance and ductility of CSPWs under monotonic loading, and the effect of asymmetric diagonal stiffeners is more significant than that of cross stiffeners. Due to asymmetric diagonal stiffeners, the maximum out-of-plane deformation of ASW is 62.02% and 39.56% of CSW and USW, respectively.(3)Under cyclic loading, compared to CSW and USW, the ultimate shear resistance, energy dissipation capacity and ductility of ASW are larger, the amplitude of stiffness degradation is smaller, and the maximum out-of-plane deformation of ASW is 82.7% and 67.2% of CSW and USW, respectively.(4)Under atmospheric corrosion, USW is least affected by corrosion, but compared to corroded USW and CSW, the ultimate shear resistance and energy dissipation capacity of corroded ASW are larger. It is revealed that asymmetric cross stiffeners are still more effective than cross stiffeners in inhibiting buckling and improving the seismic performance of corroded CSPWs. Meanwhile, the fitting formulae of the ultimate shear resistance of corroded CSPWs are a suitable engineering design reference.

## Figures and Tables

**Figure 1 materials-15-04920-f001:**
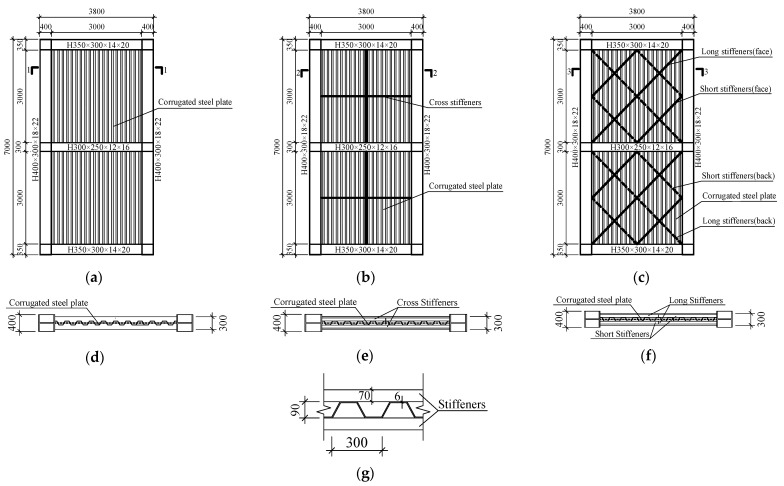
Dimension details of USW, CSW and ASW: (**a**) USW; (**b**) CSW; (**c**) ASW; (**d**) the profile of Section 1-1; (**e**) the profile of Section 2-2; (**f**) the profile of Section 3-3; (**g**) the dimensions of corrugated steel plate.

**Figure 2 materials-15-04920-f002:**
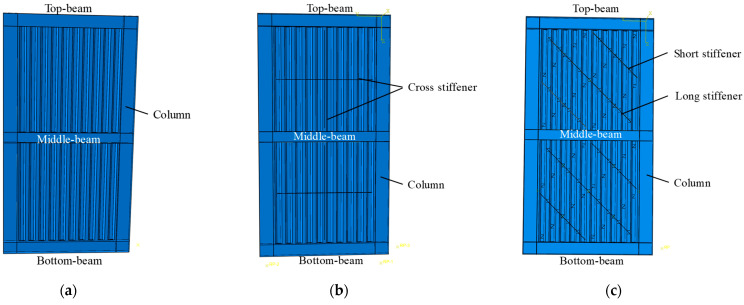
Model diagrams of USW, CSW and ASW: (**a**) USW; (**b**) CSW; (**c**) ASW.

**Figure 3 materials-15-04920-f003:**
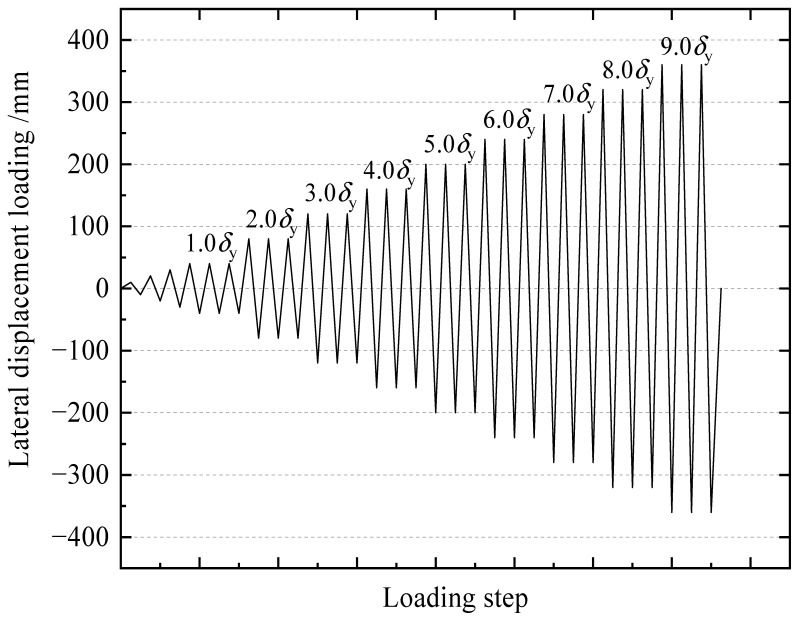
Cyclic loading of the specimens.

**Figure 4 materials-15-04920-f004:**
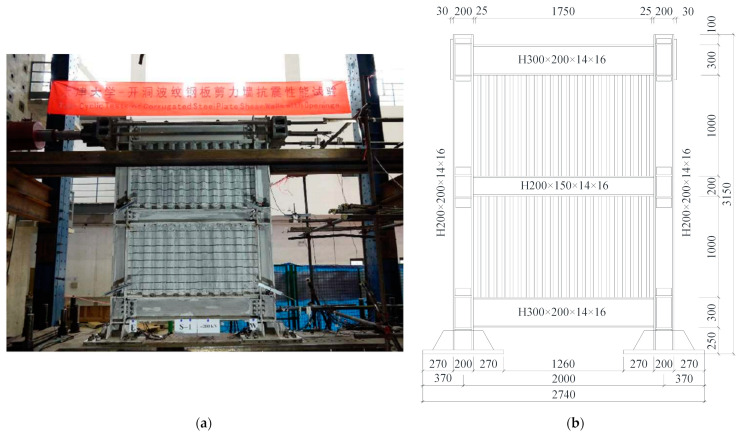
Model and sizes of the test [47]: (**a**) setup of the test; (**b**) size of the test specimen.

**Figure 5 materials-15-04920-f005:**
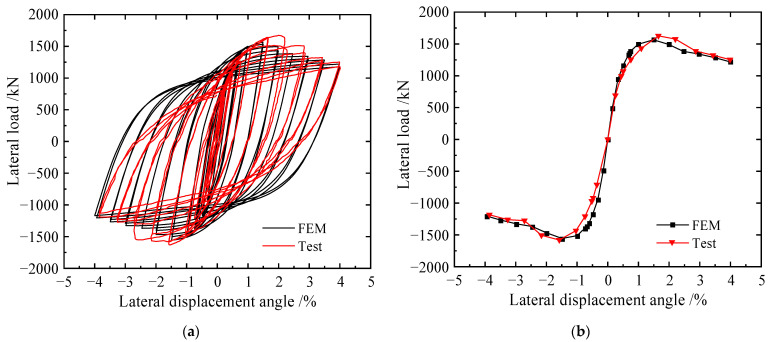
Comparisons of the load–displacement curves of specimen S-1: (**a**) hysteretic curves and (**b**) envelope curves.

**Figure 6 materials-15-04920-f006:**
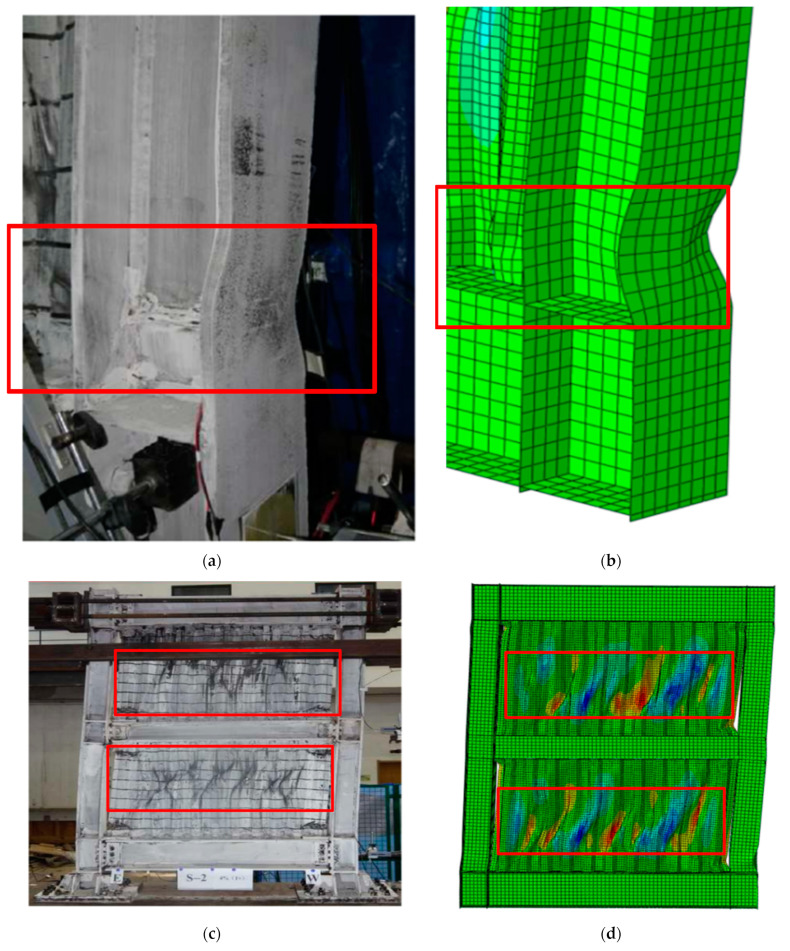
Comparison of the failure modes of the test observation and FEM analyses: (**a**) local buckling at the bottom of the test column; (**b**) local buckling at the bottom of the FE column; (**c**) global de-formation of the test specimen; and (**d**) global deformation of the FE specimen.

**Figure 7 materials-15-04920-f007:**
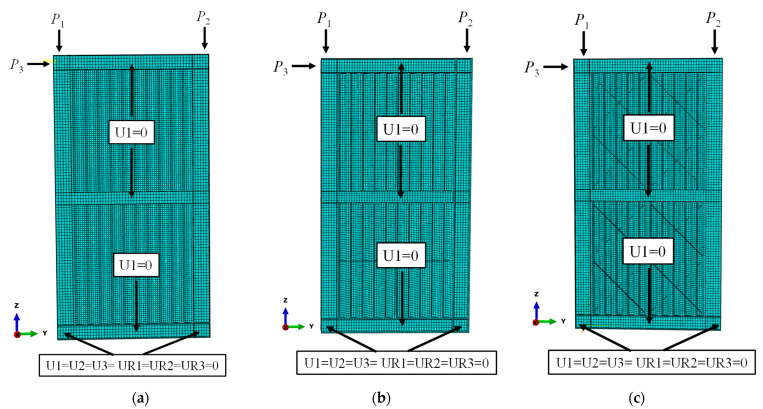
FE models of USW, CSW and ASW: (**a**) USW; (**b**) CSW; (**c**) ASW.

**Figure 8 materials-15-04920-f008:**
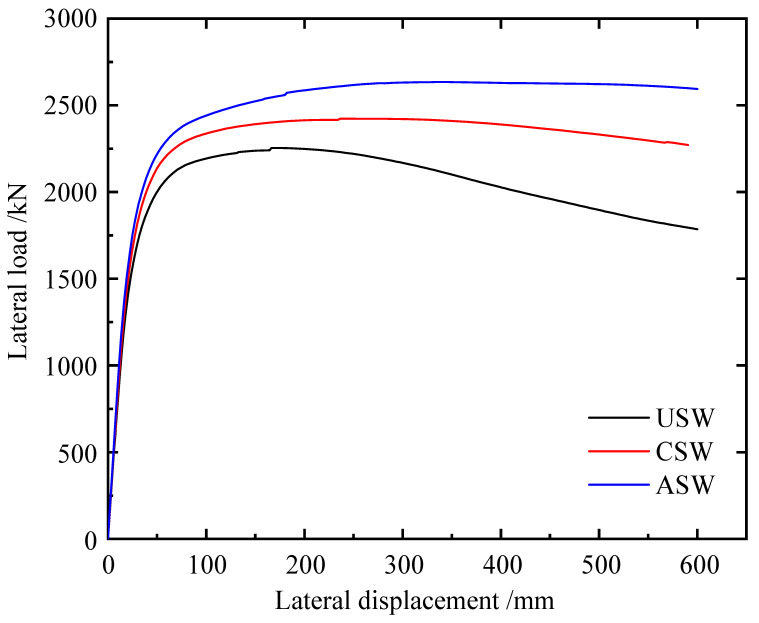
Load–displacement curves of USW, CSW and ASW.

**Figure 9 materials-15-04920-f009:**
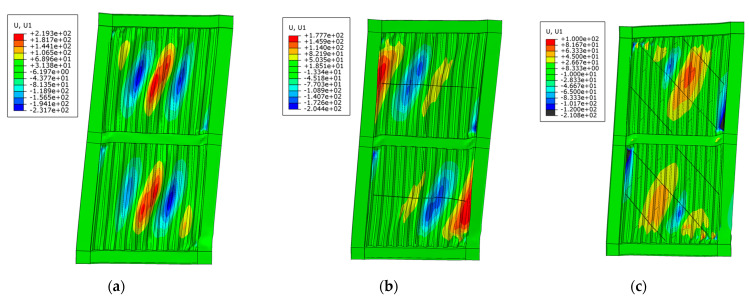
Out-of-plane deformations of USW, CSW and ASW under monotonic load: (**a**) USW; (**b**) CSW; (**c**) ASW.

**Figure 10 materials-15-04920-f010:**
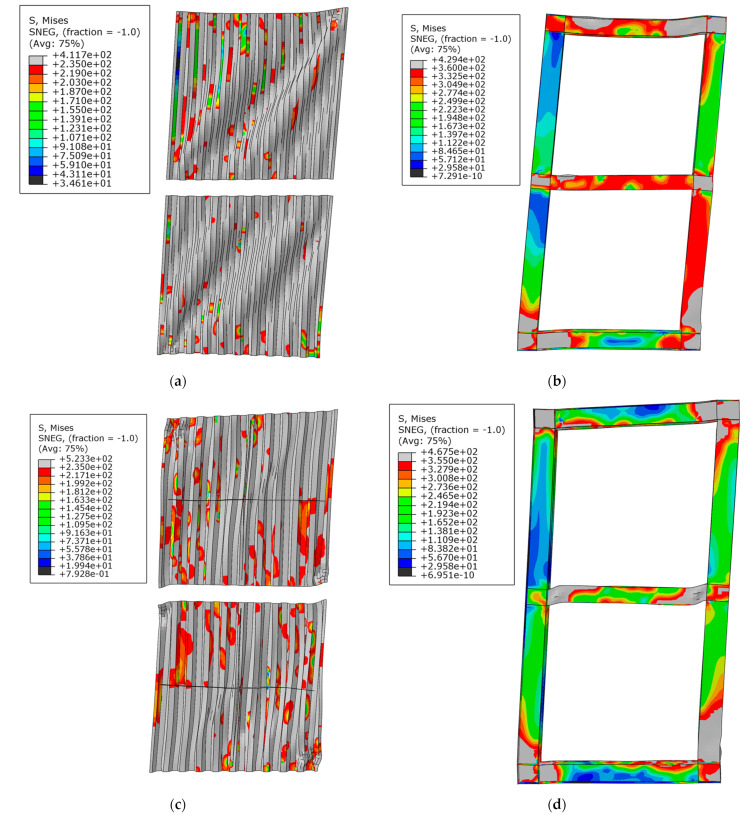
Stress distribution of damaged specimens under the monotonic load: (**a**) corrugated steel plate of USW; (**b**) boundary frame of USW; (**c**) corrugated steel plate of CSW; (**d**) boundary frame of CSW; (**e**) corrugated steel plate of ASW; (**f**) boundary frame of ASW.

**Figure 11 materials-15-04920-f011:**
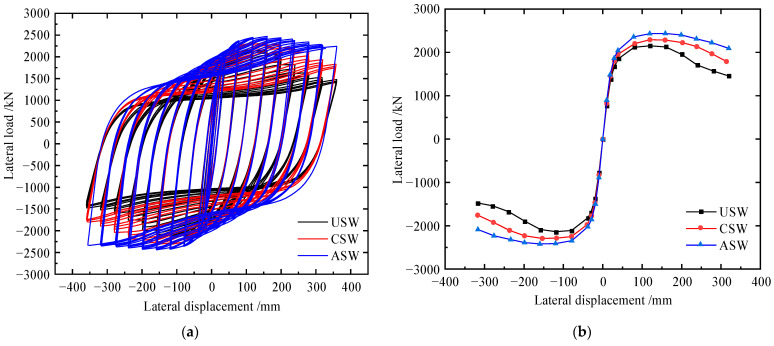
Load–displacement curves of specimens USW, CSW and ASW: (**a**) hysteretic curves; (**b**) envelope curves.

**Figure 12 materials-15-04920-f012:**
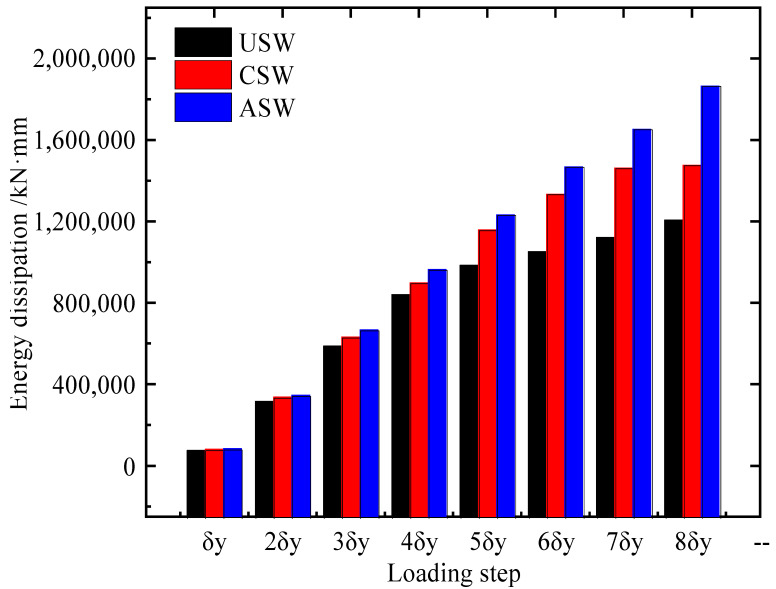
Energy dissipation of USW, CSW and ASW.

**Figure 13 materials-15-04920-f013:**
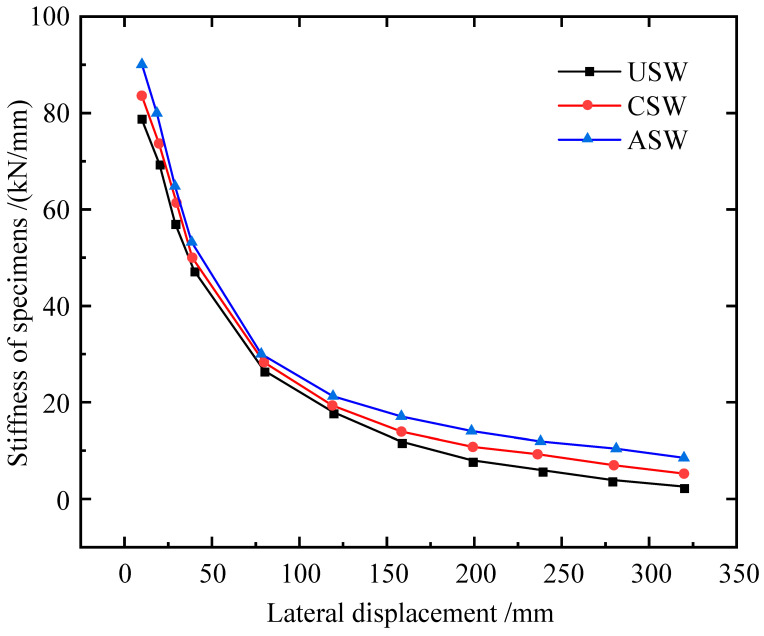
Stiffness degenerations of USW, CSW and ASW.

**Figure 14 materials-15-04920-f014:**
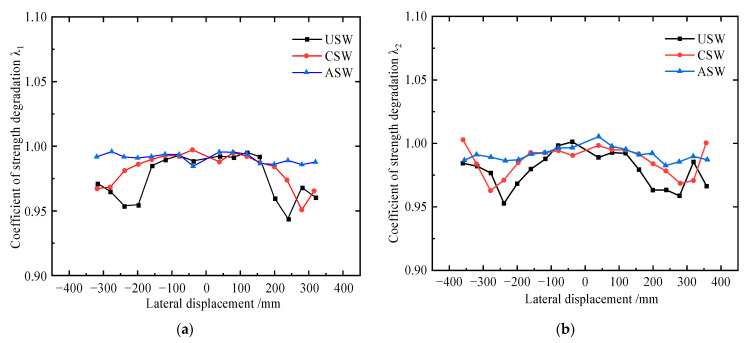
Strength degradation curves of USW, CSW and ASW: (**a**) variation curve of *λ*_1_; (**b**) variation curve of *λ*_2_.

**Figure 15 materials-15-04920-f015:**
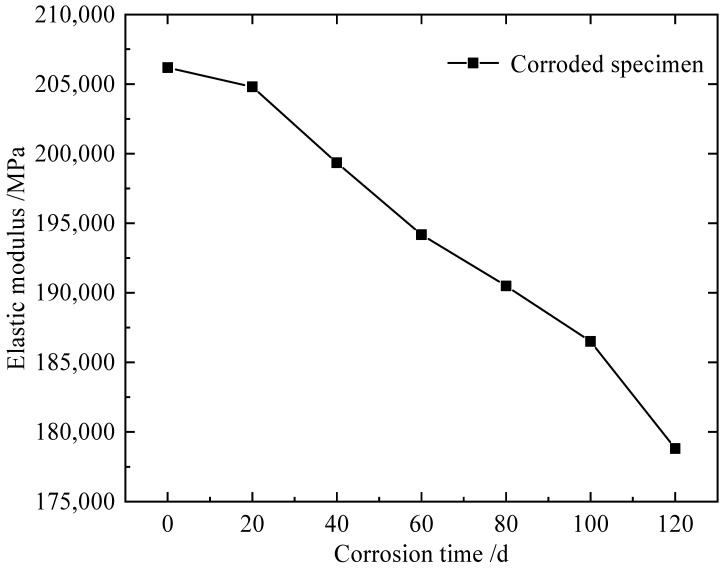
Elastic modulus of corroded specimens [35].

**Figure 16 materials-15-04920-f016:**
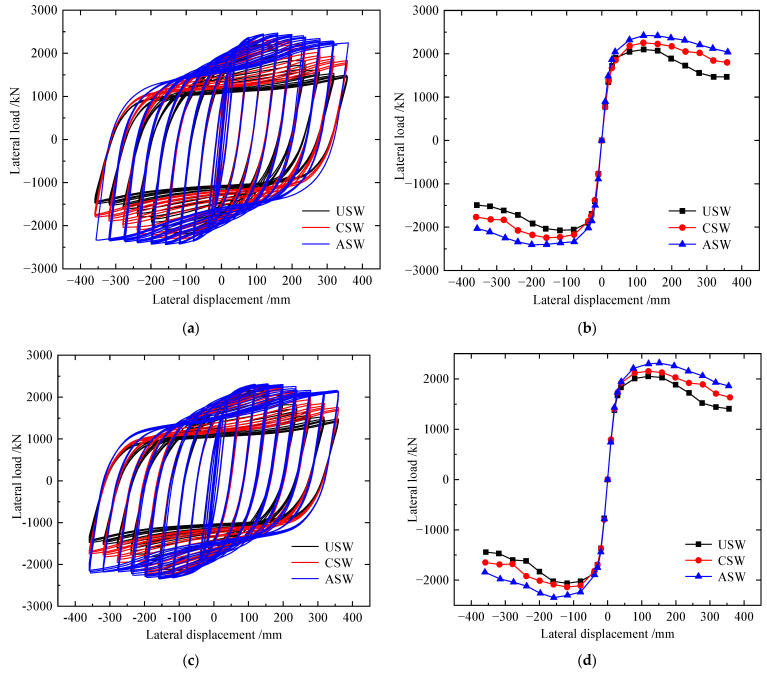
Load–displacement curves of corroded specimens: (**a**) hysteretic curves for corrosion 20 d; (**b**) envelope curves for corrosion 20 d; (**c**) hysteretic curves for corrosion 60 d; (**d**) envelope curves for corrosion 60 d; (**e**) hysteretic curves for corrosion 120 d; (**f**) envelope curves for corrosion 120 d.

**Figure 17 materials-15-04920-f017:**
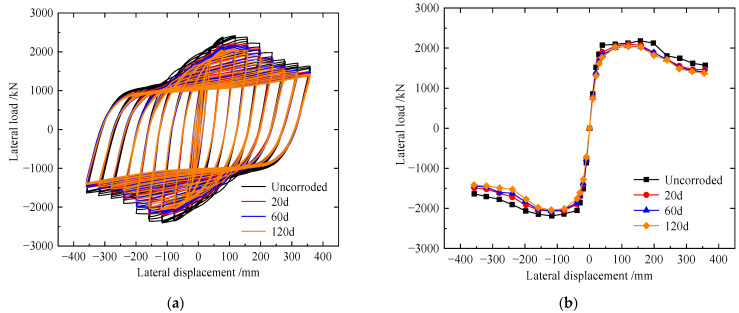
Load–displacement curves of specimens before and after corrosion: (**a**) hysteretic curves of USW; (**b**) envelope curves of USW; (**c**) hysteretic curves of CSW; (**d**) envelope curves of CSW; (**e**) hysteretic curves of ASW; (**f**) envelope curves of ASW.

**Figure 18 materials-15-04920-f018:**
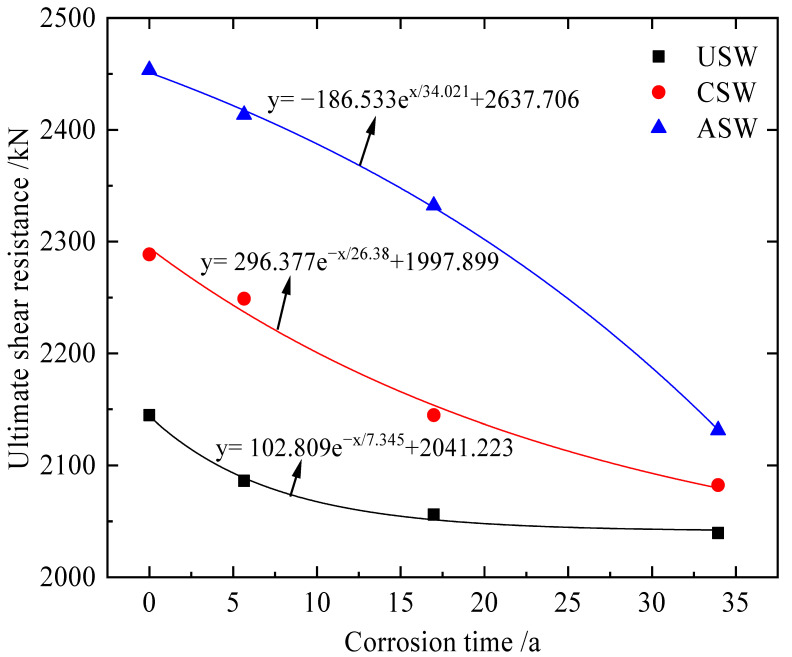
Fitting formulae of ultimate shear resistance of specimens.

**Table 1 materials-15-04920-t001:** Material properties of S-1 [47].

Items	Steel Type	Thickness *t*/mm	*E/*MPa	*f*_u_/MPa	*f*_y_/MPa	*f*_u_/*f*_y_	Elongation Ratio
H-web	Q345	14	202,558	513.3	357.3	1.44	34.6%
H-flange	Q345	16.6	183,527	468.9	334.8	1.42	34.1%
Fishplate	Q235	5.7	204,273	460	336	1.37	36.9%
Corrugated steel plate	Q235	3	193,252	451.3	312.3	1.45	39.4%

Note: *E* is the elastic modulus; *f*_y_ is the yield strength; *f*_u_ is the ultimate strength.

**Table 2 materials-15-04920-t002:** Detailed results of comparison between FEM and test.

Results	*P*_y_/kN	Δ_y_/mm	*K*_0_/(kN/mm)	*P*_m_/kN	Δ_m_/mm
Test	1300.08	20.05	99.94	1626.35	46.19
FEM	1309.99	18.28	109.04	1565.01	41.69
Error	0.76%	8.82%	9.10%	3.77%	9.76%

Note: *P*_y_ is the yield load, Δ_y_ is the yield displacement, *K*_0_ is the initial lateral stiffness, *P*_m_ is the peak load, and Δ_m_ is the displacement at *P*_m_.

**Table 3 materials-15-04920-t003:** Comparisons of the characteristic results of USW, CSW and ASW.

Specimen	*K*_0_/(kN/mm)	*P*_y_/kN	Δ_y_/mm	*P*_u_/kN	Δ_u_/mm
USW	83.02	1848.81	37.46	2254.15	170.83
CSW	88.04	1983.98	38.00	2422.66	238.89
ASW	92.50	2099.42	40.29	2633.8	334.22

Note: *P*_u_ is the maximum load, Δ_u_ is the displacement at *P*_u_.

**Table 4 materials-15-04920-t004:** Characteristic results of USW, CSW and ASW.

Specimen	*K*_0_/(kN/mm)	*P*_y_/kN	Δ_y_/mm	*P*_m_/kN	Δ_m_/mm	*μ*
USW	80.3	1792.36	36.43	2144.82	119.01	5.97
CSW	83.59	1914.53	40.6	2288.62	138.59	7.62
ASW	90.06	2001.27	42.01	2453.59	159.15	8.74

Note: *P*_m_ is the peak load, Δ_m_ is the displacement at *P*_m_, and *μ* is the ductility factor, *μ* = Δ_u_/Δ_y_.

**Table 5 materials-15-04920-t005:** Corrosion time of specimens.

Experimental Corrosion Time/d	20	40	60	80	100	120
Actual corrosion time/a	5.657	11.315	16.972	22.63	28.287	33.945

**Table 6 materials-15-04920-t006:** The elastic modulus of corroded steel [35].

Corrosion Time/d	20	60	120
*E*/Mpa	204,798	194,170	178,805

**Table 7 materials-15-04920-t007:** Characteristic results of USW, CSW and ASW for different corrosion times.

Corrosion Time/d	Specimen	*K*_0_/(kN/mm)	*P*_y_/kN	Δ_y_/mm	*P*_m_/kN	Δ_m_/mm	*μ*
20	USW	78.87	1595.78	35.72	2086.18	119.25	5.81
	CSW	80.57	1868.97	40.00	2248.94	138.59	7.28
	ASW	87.93	2001.27	41.27	2413.51	159.59	8.49
60	USW	76.98	1554.87	35.39	2056.09	119.01	5.64
	CSW	78.77	1801.25	39.43	2144.82	136.30	6.15
	ASW	81.45	1947.92	39.78	2332.26	154.77	7.71
120	USW	75.06	1542.57	35.06	2039.42	118.98	5.47
	CSW	76.46	1753.13	36.40	2082.38	135.17	6.04
	ASW	79.76	1831.91	38.14	2131.54	178.43	7.63

**Table 8 materials-15-04920-t008:** Comparison on ultimate shear resistance of FEM results and theoretical results.

Test Corrosion Time/d	Actual Corrosion Time/a	Specimen	Theoretical Results/kN	FEM Results/kN	Error
40	11.315	USW	2063.25	2071.51	0.4%
		CSW	2190.9	2232.45	1.9%
		ASW	2377.57	2395.23	0.7%
80	22.63	USW	2045.94	2052.79	0.3%
		CSW	2123.58	2110.09	0.6%
		ASW	2274.93	2298.25	1.01%

## Data Availability

Data sharing not applicable.

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
