# Peer review of "Study on the Seismic Performance of Stiffened Corrugated Steel Plate Shear Walls with Atmospheric Corrosion"

_materials, 2022, doi:10.3390/ma15144920_

Round 1

Reviewer 1 Report

This paper aims to study the seismic performance of stiffened corrugated steel 

plate shear walls with atmospheric corrosion. The overall work looks good, except some corrections need to be made, for example:

1. In Fig. 7, Why the curves are broken at a certain point? 

2. In Fig. 7, Further explanation should be added to discuss different work hardening behavior of the specimens. 

3. The discussion section is poor, the authors should describe the reason for the different properties of the specimens

Reviewer 2 Report

Authors present numerical investigation on the behavior of plane steel frames infilled with corrugated steel plate shear walls, including stiffeners, under different load conditions. Two types of stiffeners were considered and evaluated. In addition, influence of corrosion was analyzed. The FEM model is reasonably well described and validated according to the existing experimental results.

Major concern is related to the lack of corrosion effect validation. As stated by the authors:

Meanwhile, although many experimental and numerical investigations of steel structures under atmospheric corrosion have been conducted, there are still few investigations on the seismic performance of corroded CSPWs.” Please provide citations, and did you perform any type of validation of FEM model with experiment, regarding corrosion?

Please emphasize that proposed (fitted) equations are valid for analyzed parameters.

Reviewer 3 Report

The paper is dedicated to study the effect of atmospheric corrosion on the response of stiffened corrugated steel plate shear walls under cyclic and monotonic loadings. The FE software ABAQUS was used to perform parametric analysis on some configurations. The results are interesting and the context was well prepared, however, the provided draft needs further improvement before publishing. My comments are as follows:

1)     Please address in the abstract that ABAQUS finite element software (and its version also) has been used for numerical analysis.

2)    The reviewer recognizes that in the introduction, to refer to some papers, the first name was used. As an example, Alireza et al. [16] and Nima et al. [22]. It should be noted that Alireza and Nima are the first names of the Iranian authors, which should be replaced with their last names as Khaloo and Paslar, respectively. Please check this item carefully and for other papers to solve any possible mistakes.

3)    Please add a new figure to the paper (or a new sub-figure to Figure 1) to show the cross-sectional properties of USW, CSW, and ASW. This item will further help the reader to easily find all geometrical properties of the intended configurations.

4)     Mesh sensitivity analysis should be provided in section 2.1 and the dimensions of the used shell element for numerical analysis should be addressed.

5)     Figures 3.a, 5, 6, 8, 9 provided in the manuscript are of poor quality. Please replace them with high-resolution figures in the revised manuscript.

6)     What value is considered for the axial load applied to the samples? Also how this load was applied to the models to prevent recognizing it as an impact load with software?

7)     In Figure 13.b, the provided diagram for ASW is incomplete and does not contain some of the data examined in other models. Please check.

8)     The provided formulas 3-5 are really interesting. However, the proposed formulas should be validated with other examples that are not included in the fitting process. I suggest the authors to perform FE analysis with different corrosion time/a values like 12 and 25, and then compare the obtained results of FE analysis with those that will be found with the proposed formulas. The results of this analysis will help the authors to have more confidence in the proposed formulas, and the results can be presented in a new table in the revised manuscript.

9)    What load pattern was implemented in the FE analysis to perform the cyclic behavior? Please clarify and even provide a new diagram in the revised manuscript to show the considered pattern load.

Round 2

Reviewer 1 Report

The paper can be published.

Reviewer 3 Report

All my comments have been addressed in the revised manuscript. So, the paper in its present form is worthwhile to publish.